# Characterisation of the spectrum and genetic dependence of collateral mutations induced by translesion DNA synthesis

**Ádám Póti, Bernadett Szikriszt, Judit Zsuzsanna Gervai**[ID]**, Dan Chen, Dávid Szüts**[ID]*

Institute of Enzymology, Research Centre for Natural Sciences, Budapest, Hungary

* szuts.david@ttk.hu

## Abstract

Translesion DNA synthesis (TLS) is a fundamental damage bypass pathway that utilises specialised polymerases with relaxed template specificity to achieve replication through damaged DNA. Misinsertions by low fidelity TLS polymerases may introduce additional mutations on undamaged DNA near the original lesion site, which we termed collateral mutations. In this study, we used whole genome sequencing datasets of chicken DT40 and several human cell lines to obtain evidence for collateral mutagenesis in higher eukaryotes. We found that cisplatin and UVC radiation frequently induce close mutation pairs within 25 base pairs that consist of an adduct-associated primary and a downstream collateral mutation, and genetically linked their formation to TLS activity involving PCNA ubiquitylation and polymerase $\kappa$. PCNA ubiquitylation was also indispensable for close mutation pairs observed amongst spontaneously arising base substitutions in cell lines with disrupted homologous recombination. Collateral mutation pairs were also found in melanoma genomes with evidence of UV exposure. We showed that collateral mutations frequently copy the upstream base, and extracted a base substitution signature that describes collateral mutagenesis in the presented dataset regardless of the primary mutagenic process. Using this mutation signature, we showed that collateral mutagenesis creates approximately 10–20% of non-paired substitutions as well, underscoring the importance of the process.

## Author summary

DNA base substitutions are the most common form of genomic mutations, formed both spontaneously and in response to environmental mutagens. One of the main mechanisms of base substitution mutagenesis is translesion synthesis, a process that relies on specialised DNA polymerases to replicate damaged DNA templates. In addition to incorrect base insertions at the site of lesions in the template, translesion polymerases may also generate 'collateral' mutations away from the lesion due to their lower accuracy in selecting the correct incoming nucleotide. In this study, we surveyed the whole genome sequence of experimental cell clones to examine the extent and genetic dependence of collateral mutagenesis in higher eukaryotes. Looking for close mutation pairs, we found that

**Data Availability Statement:** All aligned sequence data files are available from the European Nucleotide Archive database (accession numbers PRJEB12944, PRJEB13620, PRJEB28820,

PRJEB33877, PRJEB36568, PRJEB39852, PRJEB44196, PRJEB44262). Corresponding ENA accession numbers for each sample can be found in Table J in S1 Text. Scripts used to analyse and visualise mutation data are deposited at https://github.com/szutsgroup/collateral_mutations.

**Funding:** This work was supported by the Hungarian Academy of Sciences (https://mta.hu) [Momentum Grant LP2011-015 to DS] and the National Research, Technology and Innovation Fund of Hungary (https://nkfih.gov.hu) [K_124881, K_134779, FIEK_16-1-2016-0005 and VEKOP-2.3.3-15-2017-00014 to DS, PD_121381 to BS and DS]. The funders had no role in study design, data collection and analysis, decision to publish, or preparation of the manuscript.

**Competing interests:** The authors have declared that no competing interests exist.

collateral mutations frequently occur near primary lesions generated by cisplatin or ultraviolet radiation in chicken and human cells, but are restricted to a short distance of approximately 25 base pairs. By analysing their sequence context, we showed that collateral mutations can also occur near correctly bypassed primary lesions and may be responsible for a considerable proportion of all base substitution mutations.

## Introduction

Translesion polymerases are specialised enzymes participating in the translesion synthesis (TLS) branch of the DNA damage bypass pathway, ensuring timely and smooth duplication of even severely damaged genomic DNA by locally replacing the high-fidelity replicative polymerases [1]. Due to structural properties such as missing exonuclease domains and spacious substrate binding sites [2–4], TLS polymerases are adept at replicating through DNA lesions that would otherwise stall replication forks, potentially leading to fork collapse and double stranded DNA breaks [5]. However, TLS often accomplishes this erroneously, by inducing point mutations opposite the original lesion. Indeed, well-known exogenously induced mutagenic patterns, like UVC radiation- or cisplatin treatment-associated mutagenesis have been linked to the activity of translesion polymerases [6,7].

The structural properties that facilitate translesion replication also lead to reduced fidelity on undamaged DNA [8]. Even though post-translational modifications of PCNA at lysine 164 [9,10] and other regulatory factors [11–13] limit their access to the duplicating genome, TLS polymerases probably still replicate stretches of undamaged DNA. This may primarily take place in the direct vicinity of bypassed lesions, either before switching back to replicative polymerases at the replication fork or during the filling of post-replicative gaps. This idea was first proposed in 1987 in terms of post-replicative repair in *Escherichia coli* [14], called there "hitchhiking" or "untargeted" mutagenesis, often causing clustered mutations. Since then, this phenomenon has been investigated mostly *in vitro*, or using bacterial and yeast models. The aim of this study was to corroborate these results in higher eukaryotes, and to systematically explore the associated lesion types and genetic components.

Previously, reporter gene mutation rates in *E. coli* were used to show the contribution of polymerase III to UV-associated but untargeted mutagenesis [14], and an *in vitro* gap-filling reaction demonstrated the role of polymerase V, or *umuC*, in SOS untargeted mutagenesis [15]. In yeast, the role of polymerase ζ has been suggested, either by monitoring mutation frequencies around one particular lesion inserted into the genome or transfected plasmids [16], or by the low-throughput sequencing of reporter genes [17]. In vertebrate models the only evidence comes from somatic immunoglobulin mutagenesis [18,19], and recently from the analysis of an extrachromosomally replicating plasmid in human cells [20]. However, the vertebrate approaches most probably do not fully reflect natural genomic processes, as the former only examined extra mutations at the position directly adjacent to the purported lesion site in one rather specific genomic setting, while the latter experiments utilised a hypermutagenic polymerase ζ variant acting on a plasmid.

In the present work we analysed almost 250,000 point mutations that arose in the genomes of 97 experimentally derived chicken DT40 and human TK6 and DLD-1 cell clones to explore potential TLS-dependent mutagenesis in the vicinity of various genomic DNA lesions. We found an enrichment of pairs of mutations in close proximity after treatment with various exogenous mutagens or upon the disruption of homologous recombination. Following mutagenic treatments, most of these pairs consisted of a mutagen-specific primary mutation and a

downstream collateral event. The dependence of the detected collateral mutations on PCNA ubiquitylation and polymerase κ supports the role of TLS in their formation. Notably, we confirmed the activity of the observed processes in human cells, underscoring the validity of our findings across vertebrates. Finally, we proved that a subpopulation of independent non-primary mutations are also collateral events, suggesting a fundamental role for inaccurate TLS on undamaged templates in general mutagenesis.

## Methods

### Cell culture

The following DT40 cell lines were used during the present work: wild type, $BRCA1^{-/-}$, $BRCA2^{-/-}$, $RAD51C^{-/-}$, $XRCC2^{-/-}$, $XRCC3^{-/-}$, $PALB2^{-/-}$; $MSH2^{-/-}$, $PCNA^{K164R/K164R}$, $POLH^{-/-}$, $POLK^{-/-}$, $BRCA1^{-/-} POLH^{-/-}$ and $BRCA1^{-/-} POLK^{-/-}$ (see Table A in S1 Text for their sources); $BRCA1^{-/-} PCNA^{K164R/K164R}$ cells were generated for this study by deleting exon 4 entirely and exons 3 and 5 partially in the $BRCA1$ gene using homologous gene targeting [21] in the respective single mutants. TK6 cells were obtained from the TK6 mutants consortium (http://www.nihs.go.jp/dgm/tk6.html). DLD-1 and $BRCA2^{-/-}$ DLD-1 cells were obtained from ATCC. DT40 cells were grown in RPMI-1640 medium supplemented with 3% chicken serum, 7% fetal bovine serum, 1% Pen/Strep and 50 μM β-mercaptoethanol; DLD-1 and TK6 cells were cultured in RPMI-1640 medium supplemented with 10% fetal bovine serum and 1% Pen/Strep. All cells were kept at 37˚C under 5% $CO_2$.

### Drugs and treatments

Cisplatin, methyl methanesulfonate and cyclophosphamide were obtained from Sigma-Aldrich and dissolved in water. UVC treatments were performed using a 254 nm UV lamp calibrated with a UV meter (both from UVP, Analytik Jena GmbH). Treatment conditions were as before [22–24]. Briefly, single ancestral clones were expanded from each cell line. One million cells were treated with each drug weekly for four weeks, except for cisplatin treatments of $PCNA^{K164R/K164R}$ and $BRCA1^{-/-} PCNA^{K164R/K164R}$, that were treated only twice during the same time, due to poor recovery. Treatment concentrations (10 μM cisplatin, 30 μM cyclophosphamide, 236 μM methyl methanesulfonate and 2 J/m² UVC for DT40 cells, and 6 μM cisplatin for TK6 cells) were selected according to the $IC_{50}$ of the respective drug against wild type cells. Mock treatments were also performed by maintaining cell cultures without any treatments. Single-cell descendent clones were isolated by limiting dilution 50 days (DT40 and TK6 cells) or 60 days (DLD-1 cells) after the isolation of the ancestral clones and expanded prior to DNA extraction. Genomic DNA was prepared using the Gentra Puregene Cell Kit (Qiagen).

### DNA sequencing and mutation calling

Library preparation and 2x150 nucleotide paired-end DNA sequencing was performed by Novogene (Beijing, China) using Illumina HiSeq 2500, HiSeq X and NovaSeq 6000 instruments or by BGI, (Hong Kong, China) using DNBseq (Table B in S1 Text). Sequence data preprocessing and alignment was done as before [25]. Briefly, raw sequence quality control was performed using FastQC (http://www.bioinformatics.babraham.ac.uk/projects/fastqc/), low quality and adapter sequences were removed by Trimmomatic [26], duplicated read pairs were filtered using Samblaster, alignment against the Galgal4.73 (DT40 samples) or GRCh38 (DLD-1 and TK6 samples) reference genome was done by bwa mem [27], and regions around indels were realigned using dedicated tools from GATK [28]. Substitutions and short indels were called in batches of 20–30 samples using the IsoMut tool which guarantees near-zero false

positive mutation detection, setting the filtering parameters to achieve no more than five unique SBS mutations or one indel in each ancestral clone [29] (Table C in S1 Text). To enable enhanced detection of close mutation clusters, IsoMut was modified to use samtools with the -E flag during pileup generation. To filter false positive close mutation pairs caused by mis-alignments and sequence contaminations, we applied a post-filtering step by removing mutations in regions of strong coverage fluctuations of 5 or more on both sides in a 200 bp window, and those with mapping qualities less than 40. We also verified that close pairs are on the same haplotype by showing that they were supported by the same reads.

### Analysis of external datasets

Raw whole genome sequencing data of cisplatin treated human cell lines [30] were obtained from the European Nucleotide Archive (ENA, https://www.ebi.ac.uk/ena) under accession number PRJEB21971. Steps of alignment, mutation calling, and post-filtering were done identically as described above. Whole genome sequencing alignment files of matching tumour-normal pairs of melanoma patients [31] were downloaded through the European Genome-Phenome Archive (EGA, https://www.ebi.ac.uk/ega/) using ten primary cutaneous melanoma samples representing the whole range of mutational counts from project EGAS00001001552. Bam files were remapped to GRCh38, and the close mutation-optimised version of IsoMut was ran with permissive settings to detect low allele frequency variants. Due to high mutation numbers in these samples, several post-filtering steps for near mutation pairs were applied: the average mapping quality of each mutation had to exceed 41, each mutation needed to have fewer than 2 supporting reads in any other sample, at least 3 reads had to contain both variants and the allele frequencies had to be indifferent according to Fisher's exact test.

### Statistical analyses

All data analysis steps were conducted in R. Non-negative matrix factorisation (NMF) was done using the MutationalPatterns R package [32]. Optimal NMF ranks were selected by testing multiple rank values and considering the cophenetic correlation coefficient and the residual sum of squares [33]. Differences of mutational counts were assessed by two-tailed Student's t-tests, while comparisons of ratios were tested using two-sample proportion tests or Fisher's exact tests; $p < = 0.05$ was considered significant. When assessing collateral mutagenesis *via* a polymerase slippage mechanism, events were added in two categories according to whether the mutated allele copied the previous base or not, and compared using Fisher's exact test to random base insertions, which would result in one third of the mutagenic events agreeing with the previous base. Expected close mutation ratios were obtained by analysing the intermutation ratios of as many randomly selected genomic positions as was found in the respective samples using the regioneR R package [34], however, the raw ratios in each case were divided by 2 to correct for only those close mutations that are on the same strand.

## Results

### Various mutagenic processes can generate close mutations pairs

A potential diagnostic sign of lesion bypass-associated mutagenesis on undamaged DNA may be the enrichment of close mutation pairs: one primary mutation at the original lesion, and one accompanying collateral mutation. We chose the term "collateral" over "untargeted" or "hitch-hiking" to emphasise the direct involvement of a bypassed lesion in the process. If the polymerase switch happens directly at the damage site, downstream collateral mutations should be observable in the 3' direction from lesion-coupled primary mutations if the original

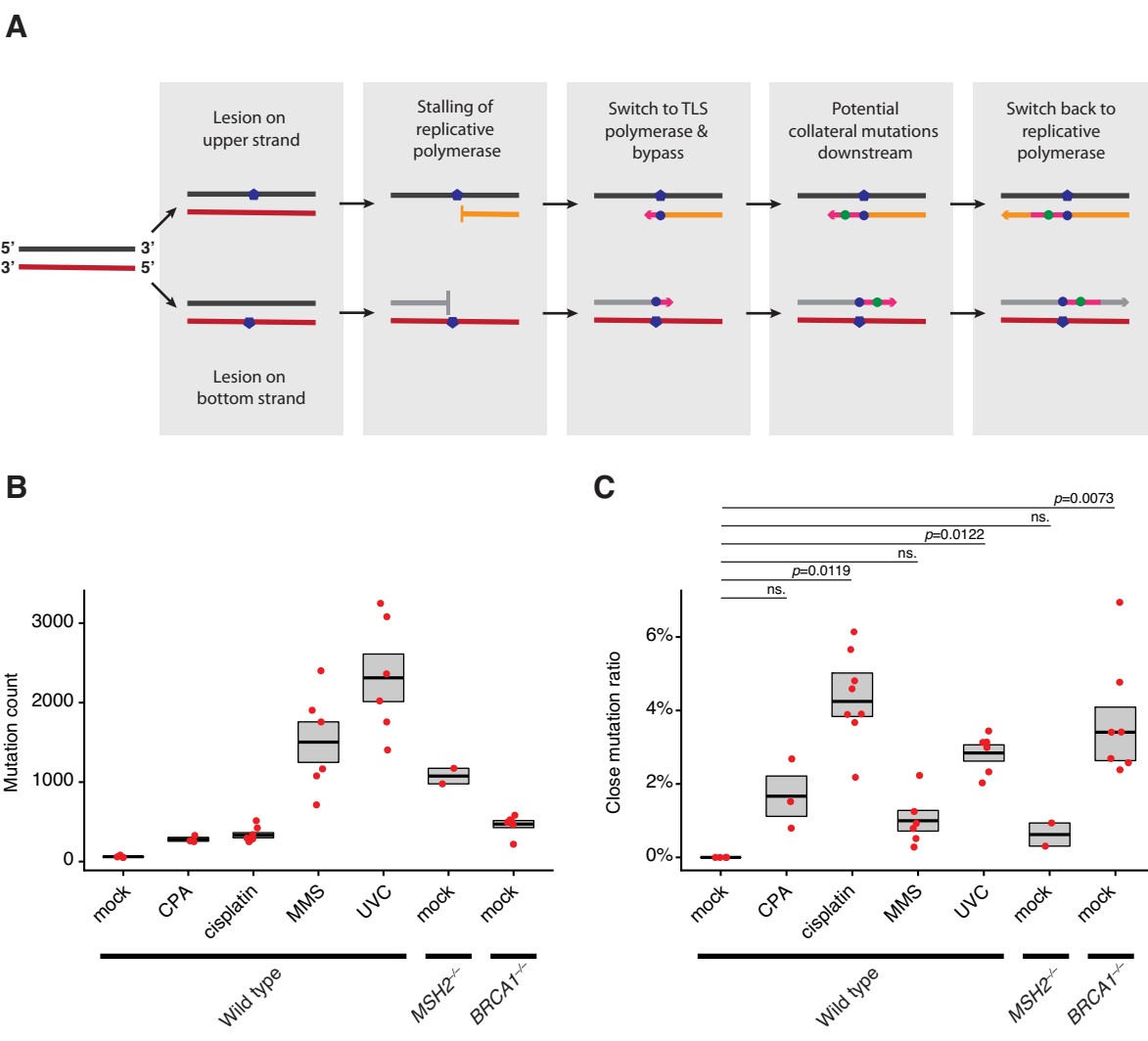

**Fig 1. Close mutation pairs are associated with specific mutagenic treatments. (A)** Model of collateral mutagenesis [14]. After the stalling of replicative polymerases (orange and grey arrows) at lesions, polymerase switch may occur. Downstream of the original lesions, TLS activity (magenta arrows) will induce collateral mutations, before switching back to replicative enzymes. Due to 5'-3' orientation of DNA polymerisation, we expect collateral mutation to the 5' if the lesion was on the upper strand (upper row), and to the 3' if the lesion was on the bottom strand (bottom row). **(B)** Single base substitution (SBS) counts after the indicated mutagenic effects. **(C)** Ratios of close mutations. Close mutations were defined as mutations with any neighbouring SBS closer than 100 bps, but without dinucleotide mutations (i.e. mutations that are directly adjacent). In (C, D) each red marker represents an independent mock or mutagen treated clone, and the box shows mean and standard error (SE).

damage site was on the lower strand, and in the 5' direction if the lesion resided on the upper strand (Fig 1A), considering that DNA sequences are usually interpreted as the 5'-3' strand of the molecule. We conducted an initial search for close mutation pairs by performing mock or mutagenic treatments on DT40 cells between two single cell cloning steps. After whole genome sequencing of ancestral and descendent clones, the newly formed mutations were identified by IsoMut, a dedicated tool for finding only unique mutations in batches of isogenic samples [29].

Close mutation pairs were explored among alterations generated by various mutagenic effects previously described in DT40 cells: *BRCA1* deficiency [23], mismatch repair (MMR) deficiency [35], cisplatin and cyclophosphamide (CPA) [22], methyl methanesulfonate (MMS)

[36] and a novel DT40 treatment for this study, UVC radiation (Tables A, B, C and D in S1 Text). Following an inspection of the distribution of intermutational distances (S1A Fig), we defined close pairs as pairs of mutations with a distance of at most 100 bps, as mutation pairs with such spacing appeared overrepresented upon certain conditions (see below). We also verified that the mutations in close pairs were on the same DNA molecule by confirming that the same set of reads supported both events in each case. We excluded double nucleotide substitutions (with intermutational distances of 1 bp), as in these cases the two mutated bases are presumably changed in the same step. Whereas all the tested mutagenic effects significantly elevated the rate of single base substitutions (SBSs) compared to mock-treated wild type cells (Fig 1B), only cisplatin, UVC treatments and the lack of BRCA1 caused significantly higher close mutation ratios (CMR), defined as the proportion of mutations in close pairs (Fig 1C), which were 4.02%, 3.01% and 3.88%, respectively. Notably, CMR was above the values expected by chance in all treated or mutant samples (S1B Fig). We also checked for close mutation pairs where one of the mutations is a short indel (1–20 bp long): we found that cisplatin, UVC and the $BRCA1^{-/-}$ genotype induced these mixed events as well (S1C and S1D Fig and Table E in S1 Text). A comparison to untreated wild type cells was not informative due to the very low event counts in those genomes, but the proportion of indels with nearby SBS mutations was significantly higher upon cisplatin or UV treatment than in the case of MMS, mirroring the data from base substitution CMRs (see S1C and S1D Fig for details).

## Exogenous mutagens cause similar patterns of collateral mutagenesis

Both cisplatin and UVC radiation were found to induce close mutation pairs: in fact, these mutagens primarily cause conceptually similar intrastrand crosslink lesions [37,38]. The majority of cisplatin-induced lesions are intrastrand crosslinks at GG or AG motifs [39,40], and it has been demonstrated that cisplatin-generated crosslinks are directly linked to induced mutations [41]. Accordingly, cisplatin-coupled mutations were found to be centred on GG or AG sequences [42,43], generating specific sparse mutational spectra, and we have characterised in detail the cisplatin-induced mutation spectrum in DT40 cells, which is dominated by N[C>A]Y, N[T>A]C and C[T>A]N primary substitutions [22]. In eight clones of wild type cells treated four times with 10 μM cisplatin, 1757 out of 3106 mutations fell in these categories (Fig 2A, the indicated mutation categories are counted as primary events). Wild type DT40 cells treated with UVC radiation developed 2340 ± 747 mutations per genome (n = 6, mean ± S.D.), with a characteristic SBS spectrum (Fig 2A) that strongly resembles the UV-associated COSMIC signature SBS7b (cosine similarity = 0.894). This signature is dominated by C>T and T>A mutations at dipyrimide sites, and these can be directly linked to UV-generated DNA photoproducts [44]. Indeed, Y[C>T]N and Y[T>A]N substitutions dominated the UVC-treated DT40 SBS spectra: 9022 out of 13874 mutations belong to these primary types. The assigned primary mutation sequence classes are very rare in mock treated cells (an average of 12 and 8 per genome for cisplatin and UVC primary classes, respectively), suggesting that 94.4% (cisplatin) and 99.4% (UVC) of mutations assigned as primary on the basis of structural information and earlier studies are correctly attributed to the treatment.

Overall, 125 and 424 mutations were positioned closer than 100 bp to another one in cisplatin and UVC treated samples, respectively. These close mutations formed 59 and 209 pairs, after the omission of 2–2 clusters with three mutations. After classifying all substitutions into primary and non-primary classes according to the above criteria, we observed that close mutation pairs significantly more often consisted of one primary and one non-primary mutation than distant (intermutation distance over 100 bp) mutation pairs in case of both cisplatin and UVC (S2A Fig), in accordance with the scheme of collateral mutagenesis. Primary mutations

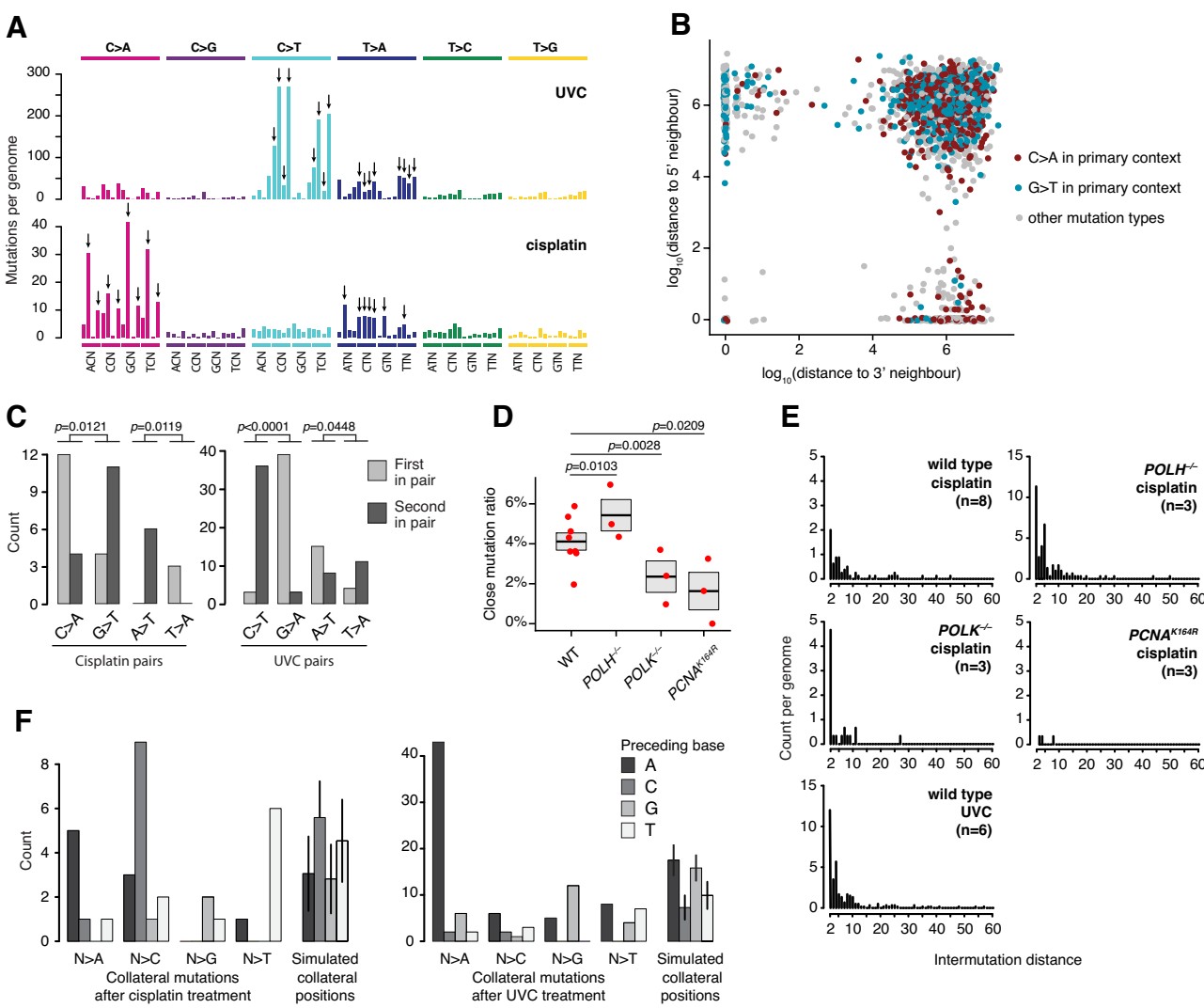

**Fig 2. Characterisation of cisplatin- and UVC-induced collateral mutagenesis. (A)** Mean triplet mutation spectra of wild type cells treated with cisplatin and UVC. Each point mutation class, as indicated by the colours and the top bars, are further classified by the identities of the preceding and the following bases, sorted alphabetically. Primary mutations arising directly at the lesion sites are marked with black arrows. **(B)** 2D rainfall plot of all cisplatin-induced mutations in wild type cells. Each point represents a mutation, coloured by its primary mutation status and identity of C>A or G>T. Position along the *x*-axis indicates the distance to the next 5' mutation in the same sample, while position along the *y*-axis indicates the distance to the nearest 3' substitution. Note the different dominant colour based on the direction of a near neighbour. **(C)** Association of primary mutation types with relative positions inside close mutation pairs in cisplatin- and UVC-treated samples. Relative position inside pairs refer to whether a given mutation is the first or the second mutation in a pair, as detected on the upper (5'-3') genomic strand. C/G>A/T, A/T>T/A mutations (in case of cisplatin) and C/G>T/A, A/T>T/A mutations (in case of UVC) are significantly enriched in the 3', or first positions of the pairs (Fisher's exact test). **(D)** Close mutation ratios of cisplatin-treated TLS mutant cell lines. Red markers show the ratios for individual clones, boxes show the means and standard errors. The significance of differences to the WT values are shown (two-sample proportion test) **(E)** Intermutation distance distributions of cisplatin- or UVC-treated wild type and cisplatin-treated TLS mutant cell lines. **(F)** Collateral mutations have a tendency to copy their immediate neighbour towards the original lesion. Collateral mutations are categorised by the variant allele (column groups) and shaded by the identity of the previous base towards the primary mutation. The rightmost categories show the distribution of previous bases for simulated collateral events around the solitary primary mutations. The simulation used the intermutational distance distributions detected in wild type cells (E) and were run 100 times with a set size equal to the number of analysed close pairs, error bars represent SD.

in pairs also showed a bias of orientation in case of both mutagens: they were significantly more often 3' members of the pairs if the lesion was on the 5'-3' strand, thus the detected mutation was G>T or A>T for cisplatin and C>T or T>A for UVC. Contrarily, they were 5'

members of the pairs if the detected mutation was the complement of the original event (C>A or T>A for cisplatin and G>A or A>T for UVC) (Figs 2B, 2C, and S2B). This is again in concordance with the hypothesis of collateral mutagenesis, as we expect collateral events downstream of the directly lesion-induced primary mutations from the aspect of the direction of DNA polymerisation. Surprisingly, this was not true in all cases, suggesting that polymerase switch and collateral mutagenesis may happen upstream of the lesion as well.

Among intermutation distances of proper collateral mutation pairs (one primary and one non-primary mutation), shorter distances were more common (Fig 2E): 86.55% and 80% of pairs were separated by at most 10 bp for cisplatin and UVC, respectively. Indeed, there were very few mutation pairs with a distance over 50 bp and almost no pairs with a distance between 100–1000 bp (Figs 2B and S2B), validating the choice of 100 bp as a limit for counting close pairs. The mutational spectrum of collateral mutations (interpreted as the spectrum of the non-primary mutations in those close pairs where the other mutation was primary) contained all base substitution types in case of both cisplatin and UVC quite evenly (S2C Fig). After considering the sequence contexts, we found that irrespective of the reference base, in cisplatin-treated samples the mutated allele had a significant tendency ($p = 0.012$, Fisher's exact test) to repeat the adjacent base towards the original lesion, i.e., the preceding base relative to the orientation of DNA replication, thus creating a dinucleotide (CC, TT, AA or GG in order of frequency, Fig 2F). This effect was also significant ($p = 3.96 \times 10^{-5}$) in UVC-irradiated samples, with a somewhat different pattern that was likely influenced by different local sequence contexts of the cisplatin- and UVC-induced lesions. Collateral mutations that copy the previous base were especially enriched in pairs with distances of no more than 5 bps (S2D Fig), suggesting multiple different sources of collateral mutations.

To explore the role of TLS polymerases in the generation of cisplatin-induced collateral mutation pairs, we exposed several mutant DT40 cell lines to identical cisplatin doses as used on wild type cells (S3A and S3B Fig) and determined CMRs (Figs 2D and S4). We were unable to perform treatments for $REV1^{-/-}$ and $REV3L^{-/-}$ mutants, as 10 µM cisplatin was lethal for these cells, and for polymerase ι, as this gene is lacking an avian ortholog. $POLH^{-/-}$ cells generated more mutations in cisplatin-associated contexts than wild type cells, and CMR was also significantly higher ($p = 0.0103$, Fisher's exact test), although the characteristics of collateral mutations were similar, except for a marked increase in the pairs with intermutation distances of 4 or 5 bps (Fig 2E). On the other hand, $PCNA^{K164R/K164R}$ cells (referred to in the following text as $PCNA^{K164R}$), expressing a non-ubiquitylable PCNA variant, generated relatively fewer cisplatin-associated substitutions among all mutations, and also showed a reduction in the number of close mutations (S4 Fig) and a significantly lower CMR ($p = 0.0012$). $POLK^{-/-}$ cells also showed significantly reduced CMR ($p = 0.0031$) (Fig 2D), though produced slightly more mutations (S3A Fig). Despite the lower CMR, the absolute number of close mutations was similar in $POLK^{-/-}$ and wild type cells (S4 Fig), but their intermutation distance profile was also different: almost all pairs in $POLK^{-/-}$ cells were separated by 2 bp (Fig 2E), and especially the mutations copying the adjacent base were missing (S3C Fig). These observations prove that TLS is indeed responsible for the observed collateral mutations, and in the case of cisplatin lesions these are partially created by Pol κ, presumably recruited by ubiquitylated PCNA.

## Exogenous mutagen-induced collateral mutations are present in human samples

We sought to validate our findings in human biological systems: cell lines and tumour datasets. Regarding cisplatin, human MCF10A and HepG2 cells were shown to generate a mutational spectrum dominated by C[C>T]N substitutions [30,45], whereas we detected a cisplatin-

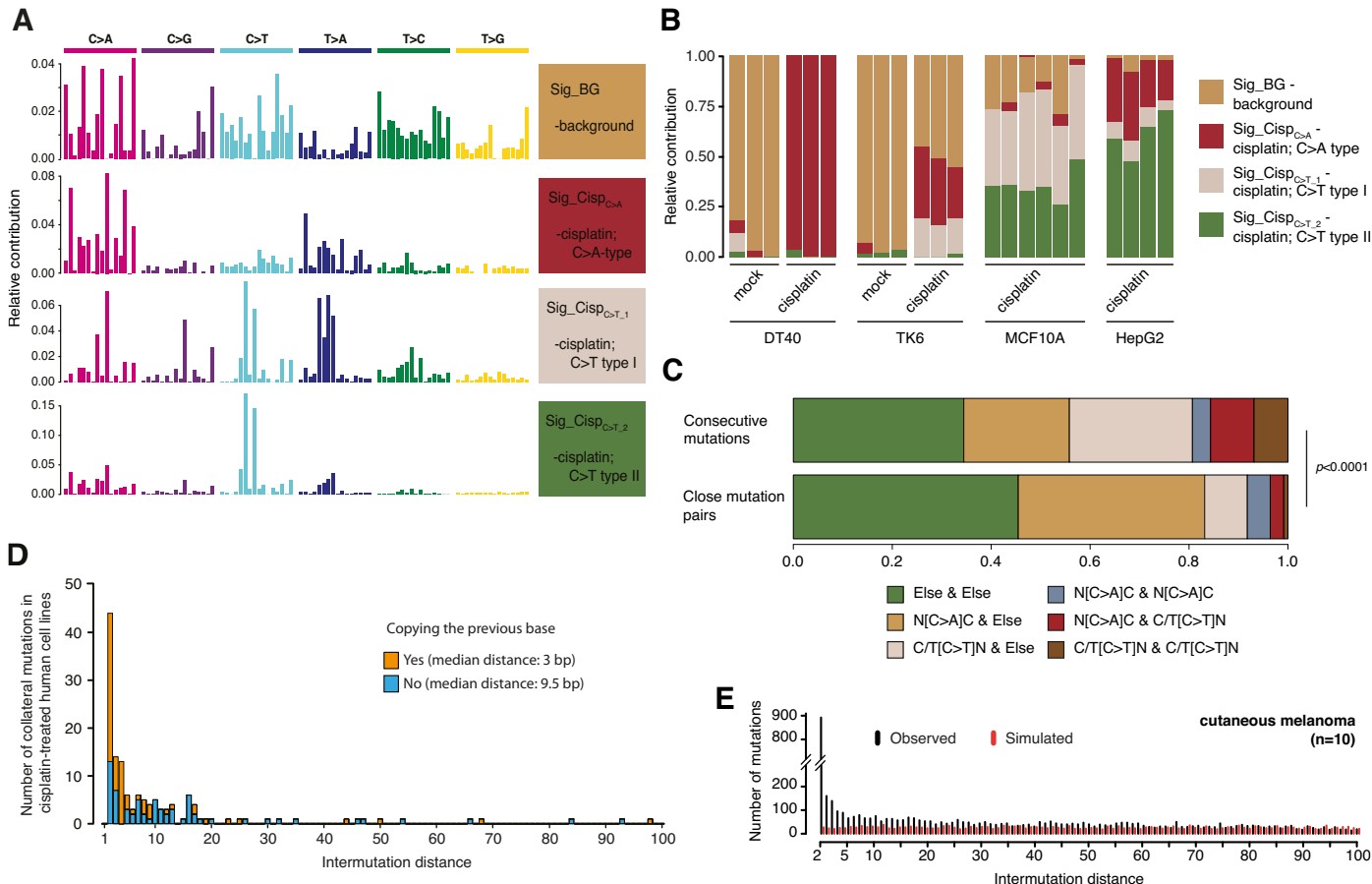

**Fig 3. Collateral mutations in human cells exposed to cisplatin or UVC. (A,B)** Spectra and relative contributions of components identified by *de novo* NMF on cisplatin-treated DT40 and human cell lines, with mock treatments included where available. **(C)** Distributions of different mutation category combinations among all distant mutation pairs, i.e. those that are separated by more than 100 bp (top bar) and close pairs (bottom bar). Categories were defined based on mutation classes N[C>A]C (main feature of Sig_Cisp$_{C>A}$), C/T[C>T]N (main feature of both Sig_Cisp$_{C>T\_1}$ and Sig_Cisp$_{\_C>T\_2}$), and all other substitutions. **(D)** Intermutation distances in cisplatin-treated human cell lines, categorised according to whether the variant allele of the collateral mutation is identical to the upstream base. Only mutation pairs consisting of a N[C>A]C-type primary event and another substitution are considered. **(E)** Intermutation distance distribution of all close mutation pairs in 10 selected melanoma samples (black), compared to an expected distribution generated by randomly simulating the same number of mutations for each sample (red).

induced SBS spectrum in human TK6 cells that appeared to be an intermediate between the MCF10A/HepG2 and the DT40 spectra [24]. After re-analysing published data from these four cell lines [24,30] using *de novo* non-negative matrix factorisation (NMF), we found that four components describe the observed triplet SBS spectra most appropriately. Three of these components were associated with cisplatin treatments, and their relative contributions varied across cell types (Figs 3A, 3B, and S5A and Tables F and G in S1 Text). Notably, only Sig_Cisp$_{C>A}$, the component associated with N[C>A]C mutations correlated with CMRs in human cells (S5B Fig), and accordingly, N[C>A]C-type primary mutations were more common among substitutions in close mutation pairs, in contrast to distant neighbouring mutation pairs, were C[C>T]N events were more common (Fig 3C). In fact, close mutation pairs in cisplatin-treated human cells fell into two main categories, those in which one of the mutations was N[C>A]C (37.7%), and those where neither mutation belonged to the typical cisplatin-induced classes (45.5%, Fig 3C). Intermutation distances were similar to those of DT40 cells: 73.68% of proper pairs were closer than 10 bps (S5C Fig), and mutations copying the previous base were also similarly enriched (Fig 3D).

We also aimed to corroborate UVC-related collateral mutagenesis using data from cutaneous melanoma whole genome sequences [31]. In 10 selected cutaneous primary tumour samples of the melanoma dataset, chosen to represent the range of total mutation counts, the intermutation distance distribution showed an enrichment of mutations closer than approximately 30 bp compared to a random simulation with the same number of mutations per tumour sample (Fig 3E). In each of the re-analysed samples, around 0.25% of all mutations belonged to a close mutation pair. Pairs within 10 bp were especially enriched compared to the random resampling (S6A Fig), suggesting that the high background of pairs within 100 bp is due to sunlight-associated hyper-mutagenesis. The first and second members of close pairs showed an asymmetry in their mutational spectra with primary C>T mutations preferentially in the upstream position, and this was more pronounced below distances of 10 bp (S6B and S6C Fig). This observation of collateral mutations in melanoma genomes lends weight to our findings in UVC-treated DT40 cells.

## Collateral mutations are associated with homologous recombination deficiency dependent mutagenesis

BRCA1 is a key component of homologous recombination (HR), an important pathway for double strand repair [46], replication fork protection [47] and DNA damage response [48]. The excess mutational load in $BRCA1^{-/-}$ cells follows a specific pattern of base changes, with strong correlation to COSMIC signature SBS3 [25], a hallmark phenotype of *BRCA1* or *BRCA2* mutated human cancers [45]. We found 131 (or 3.88%) out of 3372 substitutions in $BRCA1^{-/-}$ DT40 cells (n = 7) with another mutation closer than 100 bp. To investigate the relationship between elevated CMR ($p$ = 0.003 compared to WT, two-sample proportion test) and SBS3, we checked close mutation pairs in several HR mutant DT40 cell lines that show the same pattern of SBS3-dominated spontaneous mutagenesis [25]. Relative to wild type, we found a significant increase in CMRs for all tested mutant cell lines (Fig 4A), indicating that HR deficiency and SBS3 are generally associated with close mutations. We confirmed these findings in human $BRCA2^{-/-}$ DLD-1 cells (Fig 4B), which showed significantly higher CMR than an isogenic wild type control (1344/105,681 vs. 310/55,903, p<0.0001, two-sample proportion test). The observed CMR in $BRCA2^{-/-}$ DLD-1 cells was lower than in $BRCA2^{-/-}$ DT40, likely because DLD-1 cells also undergo MMR deficiency related mutagenesis [49], and MMR deficiency does not cause close mutation pairs (Fig 1B).

The intermutation distance distribution of close mutation pairs was broader in HR mutant DT40 cells than in mutagen-treated wild type cells: 36.1% vs. 80.0% and 86.6% of pairs were separated by less than 10 bp, respectively (Fig 4C). To assess if the observed HR deficiency (HRD) dependent close mutation pairs depend on TLS, we analysed CMR in several double mutant DT40 lines (Fig 4D). Although $BRCA1^{-/-}$ $PCNA^{K164R}$ cells produced a similar number of SBSs as $BRCA1^{-/-}$ cells (573 ± 75 vs. 482 ± 47 per genome, respectively), there were fewer close mutations (S4 Fig) and CMR was significantly lower (2/1719 vs. 131/3372, p < 0.0001, two-sample proportion test). Other investigated TLS double mutants ($BRCA1^{-/-}$ $POLH^{-/-}$ and $BRCA1^{-/-}$ $POLK^{-/-}$) showed similar close mutation numbers and CMRs to $BRCA1^{-/-}$, implying neither Pol η nor Pol κ is solely responsible for SBS3-associated close mutation pairs. The differences of intermutation distance profiles and Pol κ dependence between exogenous mutagen- and SBS3-dependent collateral mutagenesis raises the possibility of multiple parallel pathways. To investigate this issue, we treated $BRCA1^{-/-}$ single- and TLS double mutant cells with cisplatin and analysed close mutation pairs (S4 Fig). CMRs of cisplatin-treated double mutant cell lines support the idea of multimodal collateral mutagenesis: while cisplatin-treated

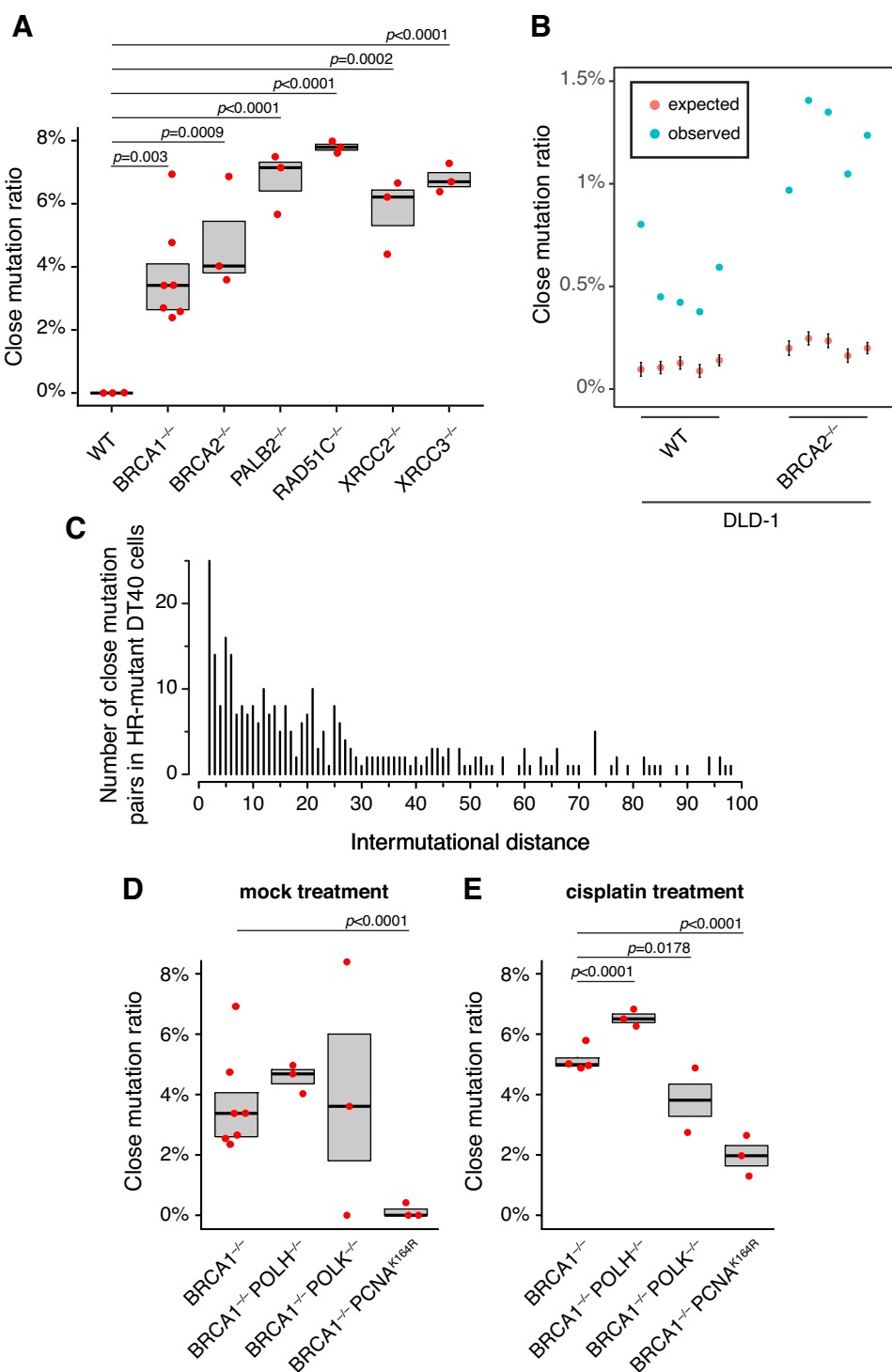

**Fig 4. Characterisation of homologous recombination-dependent collateral mutagenesis. (A)** Close mutation ratios in HR mutant DT40 cell lines are elevated, compared to the proportion in wild type cells. Red markers refer to values of individual samples, while the boxes indicate means and standard errors. Statistical significances were obtained using two-sample proportion tests. **(B)** Close mutation ratios in wild type and *BRCA2*$^{-/-}$ DLD-1 cells. Blue markers show the observed close mutation ratios for each sample; red markers indicate expected ratios drawn from 1000 random simulations of the same number of substitutions. **(C)** Intermutation distance distribution of close mutation pairs from all HR mutant DT40 clones. **(D)** Close mutation ratios in mock treated *BRCA1* double mutant cell

lines, showing that PCNA ubiquitylation is required for HRD-dependent collateral mutagenesis. **(E)** Close mutation ratios in cisplatin treated *BRCA1* double mutant cell lines.

$BRCA1^{-/-}$ $PCNA^{K164R}$ cells generated almost no close pairs, $BRCA1^{-/-}$ $POLK^{-/-}$ cells produced an intermediate level (Fig 4E).

## Collateral mutagenesis is a major contributor to the overall mutational burden

Although cisplatin-induced SBSs in DT40 are mainly lesion-associated primary mutations, all other substitution types are also induced after cisplatin and UVC treatments (Fig 2A), and HRD-dependent mutagenesis also generates all classes of point mutations. One can imagine that at least some of these broad-spectrum mutations are collateral mutations where the original lesion was bypassed correctly (Fig 5A). To investigate this possibility, we tested whether potential target sites of cisplatin-induced intrastrand crosslinks are enriched in the vicinity of 962 non-primary, solitary mutations (i.e., substitutions that do not belong to triplet mutation categories N[C>C]C, C[C>T]N or C[T>A]N, and are further than 100 bp from any other events) in all cisplatin-treated wild type samples. We found an increased count of CC and GG motifs around these substitutions, in accordance with the directionality of collateral mutagenesis (Fig 5B). The median frequency of CC or GG dinucleotides in a 300 bp wide window around all 962 non-primary, solitary mutations was 4.57% per position each, but we found within the first 10 positions 1014 CC dinucleotides to the 5' direction or GG motifs to the 3' direction, which is significantly more than the expected 880 events ($p = 0.0013$, Fisher's exact test), meaning on average 16.75 extra such events per genome in the 8 sequenced samples. On average, we found in each cisplatin treated wild type DT40 sample 3.9 proper collateral pairs among the 219.7 primary mutations, thus 16.75 extra collateral mutations would mean that 943.6 "invisible" lesion sites, or 81% of all GG lesions, were bypassed correctly. On the other hand, it also means that collateral mutagenesis is a more important contributor to the overall mutational burden: 3.9 + 16.75 mutations would account for 12.25% of the average 168.6 non-primary mutations. Similar directionally determined increases of CC and GG dinucleotide frequency could be observed in cisplatin-challenged MCF10A and HepG2 cells (S7A and S7B Fig).

Due to the tendency of collateral mutation to copy the previous base, some characteristic peaks in the triplet spectra of collateral mutations (e.g. N[T>G]G or A[T>A]N) were overrepresented (S8 Fig). We wondered whether it was possible to define a collateral mutation–specific *de novo* NMF component in the mutational spectra. NMF decomposition was performed on a set of samples including several cisplatin- and UVC-treated or HR-deficient DT40 samples, but we split the original mutational catalogues into solitary mutations for each sample, and mutations in close pairs were collected from each genotype-treatment combination. We found that five components reconstructed the original spectra most efficiently (Fig 5C and 5D and Tables H and I in S1 Text). Sig_Cisp_2 showed high similarity to Sig_Cisp$_{C>A}$ from the NMF on mixed cell types presented above (cosine similarity: 0.923), and both signatures dominated the mutation set of cisplatin-treated WT cells, illustrating the stability of NMF-based mutational signature derivation. One of these components, termed Sig_CM, was found to be associated with collateral mutagenesis, as it contributed to approximately 50% of the spectra of mutation from close pairs after each treatment (Fig 5D). The other half of the spectrum of mutations from close pairs was explained by the respective primary components for the given group, indicating that Sig_CM must be a good representation of collateral mutations in this

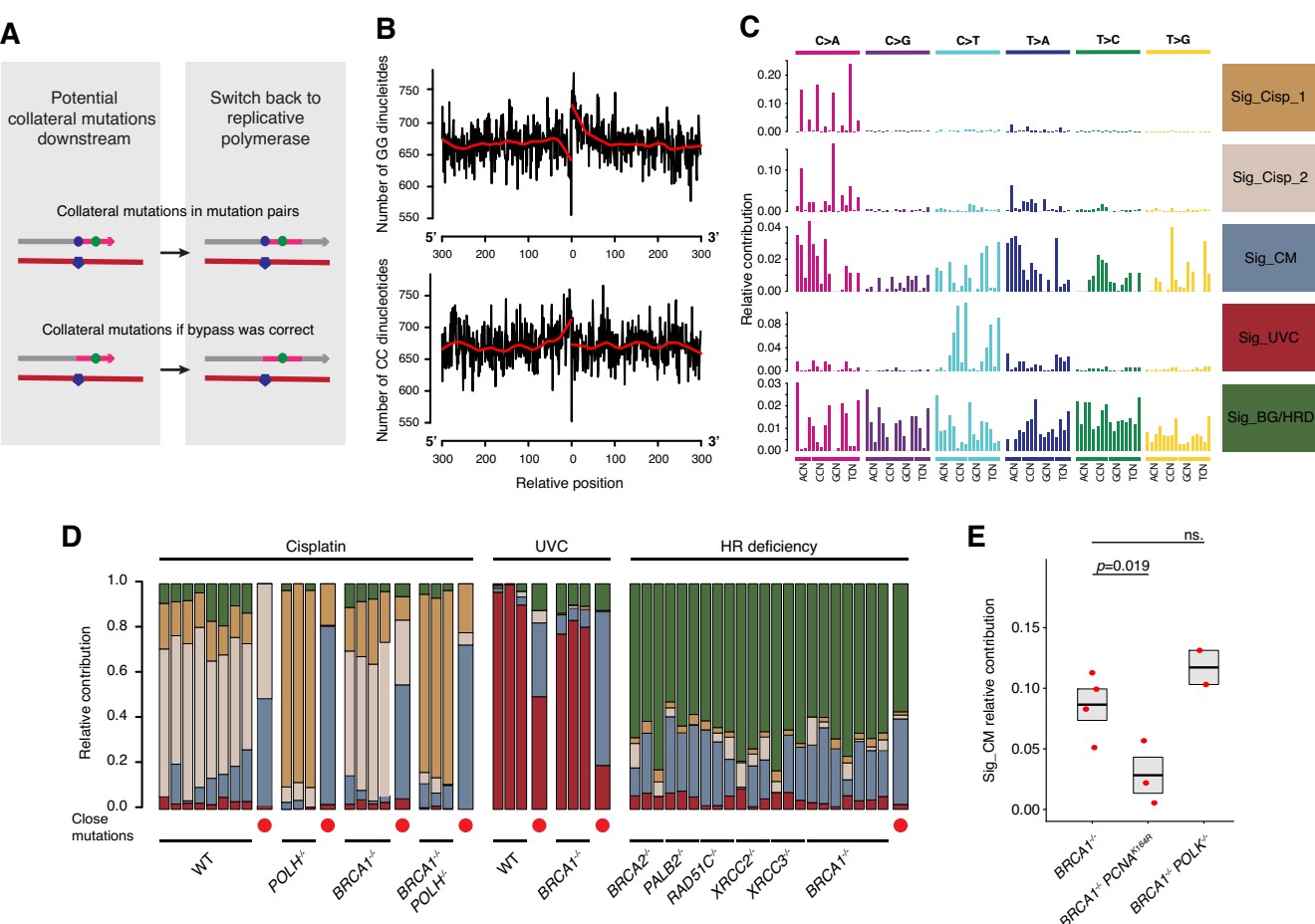

**Fig 5. Collateral mutagenesis greatly contributes to overall mutational burden. (A)** Scheme of solitary collateral mutations. Collateral mutations can occur in pairs with primary events during erroneous bypass (upper row) or downstream of correctly bypassed lesions (bottom row). **(B)** Occurrences of CC and GG dinucleotides around non-primary, solitary mutations in all cisplatin treated DT40 samples show an overrepresentation of potential lesion sites. The black lines connect dinucleotide counts in each position in a +/- 300 bps window, with lowess smoothing indicated in red. **(C)** The five components identified by *de novo* NMF on cisplatin-treated, UVC-treated and HR-deficient DT40 samples with mutations in close pairs treated as separate samples. **(D)** Relative contributions of the components in panel (C). Mutations in close pairs, denoted by red dots and wider columns, were separated in each genotype-treatment interaction, and treated as a distinct sample. **(E)** Contribution of Sig_CM to the mutation sets of cisplatin-treated samples of the indicated genotypes deconstructed using Sig_Cisp_1, Sig_Cisp_2, Sig_CM and Sig_BG/HRD. Statistical significance is indicated (two-sided unpaired *t*-test).

experimental system. Sig_CM was predicted to be present among solitary mutations as well: its absolute contribution among solitary mutations of cisplatin-treated wild type cells was 22.56 ± 6.62 (mean ± SEM), in a remarkably close agreement with the value of 17 from the CC/GG enrichment approach (Fig 5B). Sig_CM also explained ~20% of all mutations in mock treated HRD genomes, suggesting that collateral mutagenesis makes a significant contribution to genomic instability in HR deficient cells. To experimentally validate this signature, we deconstructed the mutation sets of samples that were not used during its establishment and contained many mutations: cisplatin-treated $BRCA1^{-/-}$ $PCNA^{K164R}$ and $BRCA1^{-/-}$ $POLK^{-/-}$ cells. The contribution of Sig_CM was significantly lower in $BRCA1^{-/-}PCNA^{K164R}$ than in $BRCA1^{-/-}$ controls (Fig 5E, $p = 0.019$, *t*-test), confirming the role of PCNA ubiquitylation in collateral mutagenesis. The inactivation of *POLK* did not cancel the Sig_CM mutations, probably because Pol κ appeared to have no role in HR deficiency related collateral mutagenesis (Fig 4D) and thus only a minor role in overall collateral mutagenesis in cisplatin-treated $BRCA1^{-/-}$ cells (Fig 4E).

## Discussion

Most mutations in genomic DNA appear to be generated independently of each other, with notable exceptions. In cancer genomes, mutation clusters generated by the APOBEC family of proteins have been described [50], and several groups have shown that large scale datasets of human germline variants [51] and *de novo* mutations [52] contain more mutations closer to each other than expected by chance. Multinucleotide mutational events has been described in a range of eukaryotic species [53]. TLS has been suggested to generate such mutation groups, for example with the involvement of Pol ζ in yeast [16,17], but in higher eukaryotes scarce data is available [20]. In the present work we could mechanistically show that in chicken DT40 cells, several extrinsic and intrinsic mutagenic processes (cisplatin, UVC radiation and HR deficiency) generate mutation pairs that are closer to each other than expected. Importantly, this is not the case for all mutagenic effects: for example, mismatch repair deficiency, while inducing a large number of substitutions, does not generate close pairs.

For cisplatin and UVC, the original lesion and the directly resulting substitution pattern was already known, so we could differentiate primary and collateral mutations for nearly all close pairs. Importantly, we took a conservative approach in selecting primary mutations, and only those substitutions were considered primary events that concurred with the established patterns of cisplatin- and UVC-induced lesions. Most pairs consisted of a primary and a collateral mutation, and primary mutations on the upper strand were usually the 3' members in the pair, while the ones on the lower strand were the 5' members. This relative positioning inside pairs implies that switching from replicative to translesion polymerases usually happens directly at the lesion, and the collateral mutation was generated downstream (Fig 1A). However, ~20% of events had reverse orientation, raising the possibility that polymerase switch and collateral mutagenesis can also occur upstream of the lesion, for example, if bypass takes place inside post-replicative gaps [54] or in gaps during lagging strand synthesis. In fact, upstream untargeted mutagenesis has already been demonstrated in *E. coli* [55]. The mutational spectrum of collateral events, while quite uniform by substitution type, showed a specific pattern: the altered allele was most often copying the adjacent base towards the original lesion, suggesting that the mutations arise through polymerase slippage, a well-established mechanism of polymerase-associated mutagenesis [56].

By screening several TLS mutant cell lines we found that PCNA ubiquitylation is indispensable for collateral mutagenesis in the case of the tested exogenous mutagens, with Pol κ, an enzyme recruited by this post-translational signal, being at least partially responsible: except for pairs with intermutational distances of 2 bp, all collateral mutations showing slippage disappeared in *POLK*⁻/⁻ cells. This polymerase has already been implied as an extender at platinum-induced intrastrand crosslinks [57], and a slippage-based mechanism of TLS mutagenesis has been shown for Pol κ *in vitro* [58] and for REV3 in yeast [59], similar to our findings of mutating into the previous base. Importantly, there was still some residual collateral mutagenesis in *POLK*⁻/⁻ cells. Recently a hypermutator variant of REV3L has also been shown to cause additional mutations downstream of an extrachromosomally replicated BPDE-adduct [20] in a human cell line. As Pol ζ has also been shown to be recruited to the replication fork by PCNA ubiquitylation possibly through REV1 [7], redundancy between Pol ζ and κ activity would explain the almost complete abolishment of collateral mutagenesis in *PCNA*^K164R cells.

We also observed close mutation pairs in *BRCA1*⁻/⁻ and other HR gene knock-out cell lines, revealing that HRD-dependent, COSMIC SBS3-based mutagenesis is generally associated with close mutations. The direct mechanism of SBS3-related mutagenesis is unknown: the lack of identifiable lesions also mean that we cannot clearly different primary and collateral mutations

in this case. However, a tempting hypothesis is that HR regulates the homology-dependent template switching pathway, and TLS will compensate during lesion bypass if HR is compromised. The dependence of collateral mutagenesis on PCNA ubiquitylation also suggests the role of TLS. On the other hand, we could not observe the direct role of Pol κ and Pol η in HRD-related collateral mutagenesis, which either means that non-Y family polymerases (like Pol ζ) are responsible, or that other post-translational modifications of PCNA K164 have a role. Nevertheless, we could show that HRD- and exogenous mutagen-dependent collateral processes are additive, so they form at least partially different branches of collateral mutagenesis (e.g., in the case of Pol κ contribution).

Our results clearly show that collateral mutagenesis is a short-range phenomenon, suggesting that in vertebrate cells TLS reaches for no more than about 20–30 bp downstream from the primary lesion in the template. This is in contrast with observations in yeast, where mutations attributed to Pol ζ were observed up to 1 kb from the lesion [16]. This discrepancy may be caused by a different formation of post-replicative gaps, whose lengths limit the amount of accessible DNA during bypass, or by the variance in TLS polymerase processivities between yeast and vertebrates. It is also worth noting that collateral mutations within a very few bp of the lesion cannot be considered 'untargeted' in the sense of being created on undamaged DNA, as the polymerases may still be directly influenced by the lesion in these positions. Such a mechanism may be responsible for collateral mutations created by the slippage of the polymerase, as these occur predominantly within 5–10 bp of the primary mutation (Fig 3D). In conclusion, it appears that mutagenesis by TLS polymerases on undamaged templates is efficiently restricted in higher eukaryotes.

Using NMF, we identified three different mutational patterns in all cisplatin treated cell lines, each dominated by distinct mutation types. Importantly, the relative contributions of these components were different in each case, demonstrating that the mutational patterns are also determined by the cell type, possibly through differential polymerase usage. Close mutation pairs in both human and chicken cells were mostly observed near N[C>A]C-type primary mutations, implying that the same pathway is responsible for this primary substitution pattern and collateral mutations, but it is more favoured in DT40 cells than in the investigated human cell lines. We were able to corroborate UV-related collateral mutagenesis in melanoma samples. The effect was milder, possibly due to the difference of UV subtypes involved (UVC in case of DT40 treatments and UVA- and UVB-containing solar light in case of skin cancer) and differences in the resulting lesions and their consequent bypass [60]. Also, a major subpopulation of C>T mutations are the results of spontaneous deamination of cytosines inside UV-induced cyclobutane pyrimidine dimers [61], rather than TLS. Still, together with elevated close mutation ratios in DLD-1 cells upon *BRCA2* disruption, these data prove that collateral mutagenesis is not confined to DT40 cells but occurs universally in higher eukaryotes.

Using NMF and analysis of sequences surrounding non-primary, solitary mutations, we obtained evidence that collateral mutations can happen independently of primary mutations. Approximately four times more collateral events were estimated in cisplatin-treated wild type DT40 cells than found in close pairs, at least partially explaining the elevated rate of non-primary base substitutions in platinum-treated DT40 cells [24]. The extracted collateral mutation-specific triplet SBS signature appeared to be a significant component of HRD-dependent mutagenesis, suggesting that the low fidelity of TLS polymerase activity makes an important contribution to base substitutions arising in *BRCA1* or *BRCA2* deficient cells.

In summary, we have genetically characterised the TLS-dependent collateral mutagenic process in higher eukaryotes and demonstrated that both exogenously induced DNA adducts and spontaneous endogenous processes give rise to base substitutions of a similar spectrum. Depending on the mutagenicity of the bypass of the primary lesion, collateral mutagenesis

may even be the major mutagenic effect of TLS and appears to make an important contribution to base substitution mutagenesis in multiple settings.

## Supporting information

**S1 Fig.** (A) Distribution of distances to the next 3' point mutation for all single base substitutions in all analysed DT40 cell clones exposed to various mutagenic effects. (B) Expected (black dots) and observed (red markers) close mutation ratios upon all analysed mutagenic effects; expected ratios were obtained by 100 random simulations of the same number of point mutations for the given effect. (C) Short insertion and deletion counts in DT40 cells. The same mutagenic effects were analysed as in Fig 1B and 1C. Deletions and insertions are plotted separately (with blue and green colours, respectively). (D) Ratios of close mutations where one of the events was a short deletion or insertion. Statistical significances were calculated using two-sample proportion tests.
(TIF)

**S2 Fig. Further characterisation of cisplatin and UVC induced collateral mutagenesis.** (A) Consecutive mutation pairs were classified as close or distant, and mutations within the pairs were categorised as 'primary' or 'non-primary' according to Fig 2A. The proportions of possible combinations are shown. (B) 2D rainfall plot of intermutation distances in the genomes of UVC-treated wild type DT40 cells. (C) Raw spectra of collateral mutations after cisplatin and UVC treatments. (D) Average counts of collateral mutations in cisplatin- and UVC treated wild type cells with the indicated intermutational distances, coloured by whether they copy the previous base towards the original lesion. Distance values are shown between 2 and 50 bp.
(TIF)

**S3 Fig. Mutations in cisplatin-treated TLS mutant DT40 cell lines.** (A) Mutation counts in the treated genotypes. Individual observations are symbolised with red dots, the group-wise median and interquartile range are marked within the grey box. Statistical significances for differences of mutational numbers are determined by Student's t-test. (B) Triplet spectra of substitutions in the treated genotypes. (C) The relationship between intermutational distances and whether the collateral mutations copy the adjacent base in WT, POLH$^{-/-}$ and POLK$^{-/-}$ cells.
(TIF)

**S4 Fig.** The total number of SBS mutations per genome (left panels), the number of close mutations (middle panels) and the close mutation ratios (right panels) for the indicated genotypes and treatments. Close mutations were defined as mutations with any neighbouring SBS closer than 100 bps, but without dinucleotide mutations (i.e. mutations that are directly adjacent). Each red marker represents an independent mock or mutagen treated clone, and the box shows mean and standard error (SE).
(TIF)

**S5 Fig. Collateral mutations in cisplatin-treated human cell lines.** (A) Reconstruction errors of *de novo* NMF as estimated by relative RMSD: root of means of squared differences between the reconstructed and observed mutation spectra, normalised by mutational counts in the respective samples. (B) Relationship between the relative contribution of Sig_CispC>A and close mutation ratio in cisplatin treated TK6, MCF10A and HepG2 cells. (C) Intermutation distance distributions of close mutation pairs in cisplatin-treated human cells.
(TIF)

**S6 Fig. Collateral mutations are present in melanoma.** (A) Close mutation ratios in 10 selected melanoma samples. Ratios of pairs with distances between 2 and 10 bp (red) or

between 11 and 100 bp (blue) are compared to ratios obtained for the same number of random genomic positions for each sample (heavy and light shades, respectively). (B) Association of primary mutation types with relative positions inside close mutation pairs. Asymmetries are more pronounced at closer distances (right barplot) because the high mutation levels result in unrelated mutations randomly occurring close to each other. Statistical significances (p < 0.05, Fisher's exact test) are indicated with asterisks. (C) Cause of asymmetries during collateral mutagenesis. Collateral events (green dots) are expected downstream of the original lesion.
(TIF)

**S7 Fig. Collateral mutations can arise during error-free bypass.** CC and GG dinucleotide (i.e., potential cisplatin lesion site) frequencies in +/- 300 bp windows around non-primary, solitary mutations in cisplatin-treated MCF10A (A) and HepG2 (B) cells. Primary mutations were defined as N[C>A]C, C[C>T]N and C[T>A]N. The black lines connect dinucleotide counts in each position, with lowess smoothing in red.
(TIF)

**S8 Fig. SBS triplet spectra of mutations in close pairs in cisplatin treated DT40 cells.** Each spectrum belongs to the genotype indicated at the top.
(TIF)

**S1 Text. Supplementary information cited in the main text.** Table A. List of cell lines used in this study. Table B. List of sequenced samples and sequencing statistics. Table C. Catalog of all detected and post-filtered SNVs. Table D. Catalog of all close mutation pairs. Table E. Catalog of close mutation pairs containing indels. Table F. *De novo* NMF components of cisplatin treated cell lines (Fig 3A). Table G. *De novo* NMF contributions of cisplatin treated cell lines (Fig 3B). Table H. *De novo* NMF components of DT40 samples (Fig 5C). Table I. *De novo* NMF contributions of DT40 samples (Fig 5D). Table J. ENA accession numbers for archived sequencing data.
(XLSB)

**S1 Data. Source numerical data underlying the main and supplementary figures.**
(XLSX)

## Author Contributions

**Conceptualization:** Dávid Szüts.

**Data curation:** Ádám Póti.

**Formal analysis:** Ádám Póti, Dávid Szüts.

**Funding acquisition:** Dávid Szüts.

**Investigation:** Bernadett Szikriszt, Judit Zsuzsanna Gervai, Dan Chen.

**Supervision:** Dávid Szüts.

**Visualization:** Ádám Póti.

**Writing – original draft:** Ádám Póti, Dávid Szüts.

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
