## [Decision Letter · Decision Letter 0]

16 Nov 2021

Dear Dr Szüts,

Thank you very much for submitting your Research Article entitled 'Characterisation of the spectrum and genetic dependence of collateral mutations induced by translesion DNA synthesis' to PLOS Genetics.

The manuscript was fully evaluated at the editorial level and by independent peer reviewers. The reviewers appreciated the attention to an important problem, but raised some substantial concerns about the current manuscript. Based on the reviews, we will not be able to accept this version of the manuscript, but we would be willing to review a much-revised version. We cannot, of course, promise publication at that time.

Should you decide to revise the manuscript for further consideration here, your revisions should address the specific points made by each reviewer. Also please, address the comment by Reviewer 2 on the lack of the source data for values that are shown on your figures. We will also require a detailed list of your responses to the review comments and a description of the changes you have made in the manuscript.

If you decide to revise the manuscript for further consideration at PLOS Genetics, please aim to resubmit within the next 60 days, unless it will take extra time to address the concerns of the reviewers, in which case we would appreciate an expected resubmission date by email to plosgenetics@plos.org.

[LINK]

We are sorry that we cannot be more positive about your manuscript at this stage. Please do not hesitate to contact us if you have any concerns or questions.

Yours sincerely,

Dmitry A. Gordenin, Ph.D.

Associate Editor

PLOS Genetics

Gregory P. Copenhaver

Editor-in-Chief

PLOS Genetics

Reviewer's Responses to Questions

**Comments to the Authors:**

Reviewer #1: Póti et al characterized a process they refer to as collateral mutagenesis. They find that in mammalian cells, some mutagenesis processes (related to cisplatin, UVC radiation, defects in homologous recombination pathways) create not only a primary mutation but a second one close by, within 100 base pairs of the first. The primary mutation is easy to identify for these two mutagens but may be more difficult, for example for deficiencies in BRCA1. They characterized this process in several cell systems and primary tumors and identify some of the mechanisms of collateral mutagenesis such as an at least partial requirement for ubiquitinylated PCNA and DNA polymerase kappa. This is a very interesting paper and warrants publication. However, I have several comments or questions:

1) How much of a concern are sequencing artefacts for interpretation of all data? Another technical question is: Were all the mutation pairs identified on the same DNA sequencing reads?

2) What do the authors think, why MMS doesn’t produce collateral mutations? Could it be that these collateral mutations are induced by NER gap filling rather than by a DNA damage bypass polymerase? This possibility also comes to mind because most dual mutations were separated by at most 10 bp for cisplatin and UVC.

3) The authors discuss UVC-induced mutations and their similarity to the COSMIC mutation signature 7b in terms of C>T and T>A mutations. However, SBS7b (or SBS7a) have almost no mutations of the type T>A.

4) For the published data from melanoma samples, how were these samples selected (only 10 were used)?

5) The authors present no clear statistical analysis to determine if the close mutation pairs occur at frequencies greater than expected by chance.

6) That authors mention that they were unable to perform treatments for REV1–/– and REV3L–/– mutants, as 10 μM cisplatin was lethal for these cells. However, they could have used a lower dose of cisplatin.

7) In melanoma genomes, around 0.25% of the mutations belonged to the close mutation pairs. This number seems much smaller than the authors' own UVC mutation data with a cell line where those mutations seemed to be around 3% of all mutations. How do the authors explain such a large difference?

8) Regarding SBS3 in BRCA1-deficient cells, SBS3 has a rather heterogenous, almost random distribution of mutations. Could this signature be an artifact? It may be difficult to assign a specific pattern to SBS3.

9) The part on collateral mutagenesis on a presumably correctly bypassed lesion is a little bit more speculative. If the substitutions are not paired, then how do you know that this is collateral mutagenesis?

10) Figure S5B:

When the authors display the mutation type for the first in pair and second in pair mutations, they look similar. But according to the author's conclusion, the second mutation should be more of a random type. Also, does the first mutation correspond to dipyrimidine positions?

The data is not consistent with Fig. 2C.

11) Figure S6:

I don't understand this Figure. This is showing CC or GG frequencies around non-primary solitary mutations. However, there is a huge spike of GG/CC at the zero-position suggesting a primary event.

Minor points:

Fig. 2F:

"Collateral mutations tend to copy their immediate neighbour"

Please explain this a little better. Does it mean that the process would create dinucleotides of the same type, such as AA? If so, why would this be the case?

References 24-26 and possibly others:

These references are incomplete.

Reviewer #2: The manuscript “Characterization of the spectrum and genetic dependence of collateral mutations induced by translesion DNA synthesis” evaluates the production of closely-spaced mutations in chicken DT40 and human cells. The authors utilize whole genome sequencing, primarily of DT40 cells, to identify mutations associated with cisplatin treatment, UVC treatment, or homologous recombination (HR) deficiency. They determine that these treatments and HR-deficiency induce characteristic base substitution signatures. Among these mutations, they also find evidence of closely spaced mutations (i.e. multiple mutations within 100 bp of each other). Based upon the characteristic base substitution patterns of cisplatin and UV treatment, the authors determine that these mutation pairs predominantly consist of a primary mutation resulting from mutagenic bypass of a damaged base, and a secondary, downstream mutation that does not fit the characteristic signature of the mutagen and therefore is likely due to TLS synthesis making mistakes on an undamaged template. The authors confirm that these collateral mutations are caused by TLS polymerases as they are less prevalent in DT40 cells with a PCNA K164R mutant. A role for pol Kappa in generating collateral mutations in response to cisplatin is also observed. Similar patterns of collateral mutations are observed in cisplatin-treated human cancer cell lines and in UV exposed melanoma genomes, suggesting that similar mechanisms observed in DT40 cells also operate in human cells. Finally, the authors attempt to find evidence that TLS recruitment to DNA lesions can result in the secondary mutations (i.e. mutations at the damage site) during error-free bypass of a DNA lesion and estimate the contribution of these types of events to the overall mutation load.

In general, the sequencing assays and analyses are well executed and the data within the manuscript appears to support a role of TLS in generating collateral mutations during lesion bypass in chicken and human cells. Similar induction of collateral mutations by TLS in E. coli and yeast is well known. Additionally, a recent publication has shown a reduced fidelity variant of pol zeta can mediate collateral mutations during site specific bypass of a benzo[a]pyrene in a plasmid substrate in human cells. As such, the confirmation that TLS induces collateral mutations in chicken genomes exposed to DNA damaging agents is not surprising. The manuscript does, however, provide a few additional pieces of mechanistic information on which polymerases contribute to collateral mutagenesis in higher eukaryotes. I have the following specific critiques.

1) In Figs 1c, 2d, 4a, 4b, 4d, 4e, S4b, and S5a the number of close mutation pairs appears to be standardized by the total number of mutations in the sample (i.e., close mutation ratios are reported and analyzed statistically). This is potentially problematic as changes in solitary mutation numbers could impact this ratio and the statistical results. Therefore, the number of close mutation pairs per genome should be presented and statistically evaluated to determine if the occurrence of collateral mutations is changing dependent on genotype or treatment.

2) In Fig. 2F, the authors state that collateral mutagenesis during cisplatin-lesion bypass involves a slippage mechanism and copying the upstream base. This appears to be true for cisplatin-induced mutations, however, no statistics are performed. Moreover, the authors indicate that this is a general mechanism as a similar pattern is seen in UVC treated cells. However, for UVC treated cells, this appears to primarily be true for N>A mutations and is not true for N>C and N>T mutations. Without having a statistical evaluation of the similarity between the cisplatin and UV treated cells, this is an overstatement.

3) No genetic requirements for collateral mutations in UVC-treated cells are presented. This seems to miss an opportunity to confirm whether multiple TLS polymerases contribute depending on what type of lesion is causing the damage. For example, is pol eta generating collateral mutations in UVC treated cells and potentially melanomas. The data showing pol kappa does not influence collateral mutations in HR-deficient cells seems to support that differences in the initiating lesion determine with polymerase is causing the collateral mutations.

4) Similarly, REV1-/- and REV3L-/- mutants were not evaluated due to lethality with the treatments. Could lower doses of cisplatin be used to evaluate their roles in collateral mutations by this sequencing method?

5) In Fig.5, the authors assess the enrichment of CC and GG sequences near solitary mutations that do not fit the cisplatin substitution signature in cisplatin treated DT40 cells. They show a small enrichment of these sequences suggesting that some of these solitary mutations may result from TLS polymerase recruitment to a cisplatin lesion, which was bypassed error-free, and subsequently TLS made another error. Based on this, the authors try to calculate the percentage of mutations that are collateral TLS errors. They determine the number of solitary, non-cisplatin signature mutations that have a CC dinucleotide within 10 nt upstream of the mutation or a GG dinucleotide within 10 nt downstream and suggest that all of these mutations (17 in total) are collateral TLS mutations to suggest that 12% of non-primary mutations are due to this mechanism. This percentage, however, is likely a significant over-estimation. By random chance, one would expect to find a CC or GG dinucleotide within 10 nt of half of these mutations, which would place the minimal estimate more in the range of 8 or 9 events. The number of these events observed, then becomes very small and may be heavily influenced by random variations.

6) The de novo NMF-derived signatures for cisplatin-induced mutations does not appear to be very stable between the analyses in Fig. 3a and Fig. 5c. This often results from utilizing NMF on datasets with small sample sizes and likely introduces significant error in attributing specific mutations to a specific signature and identifying the cause of the signature. This is particularly problematic for trying to parse mutations that are caused by TLS polymerases making errors across from a cisplatin lesion from collateral TLS mutations that are also induced by damage which recruits the TLS polymerase. This makes me question how specific the Sig_CM signature in fig. 5 is for collateral mutations or whether it also includes cisplatin-induced mutations. An experimental validation that Sig_CM is truly a collateral mutation signature would help. Do the pol kappa-/- or PCNA K164R cells lack Sig_CM specifically or are they required for all signatures? This was not assessed in fig. 5d.

7) The basis for the author’s claim that up to 20% of mutations are collateral mutations is based on the contribution of Sig_CM in specific genomes and if this signature is not specific for only collateral mutations, then that prevalence is a significant over-estimation. I believe an estimation based on the enrichment of CC and GG sequences near solitary, non-primary mutations would be more accurate. Based on the arguments in point 5, it would then be about 3-4% of total cisplatin-induced mutations.

Minor:

1) On page 5, it is stated that cisplatin, UVC, and BRCA2-deficiency induce close mutation pairs involving one insertion/deletion mutation, but it’s not statistically significant. If it is not significant then how can they be confident that event class is induced. This statement should be removed.

2) Indicate that the PCNA-K164R cells have mutations in both alleles.

3) The trinucleotide contexts on the x-axis of fig. 2a indicate that all mutations are at C bases. The rightmost half need to be corrected to be in T bases.

4) The y-axis of fig 3d, 4c, S4c, and S5B just state “frequency”. Specify what ratio is being taken. Similarly, Fig 5b and S6 just state CC or GG frequency, but it is unclear what is being divided to generate these numbers.

5) Fig. S3 and S7 lack y-axis titles.

Reviewer #3: The authors analyze closely spaced mutation pairs in chicken DT40 cells treated with various mutagens or carrying DNA repair defects. They find that cisplatin, UVC, and homologous recombination defects elevate the frequency of closely spaced mutations. They further detect such mutation pairs in human cell lines and melanoma. The analysis is based on sequencing whole genomes of the cell lines with or without mutagenic treatment, deducing “primary” lesions from the known specificity of the mutagen, and quantifying mutations in the vicinity of the presumed primary lesions. The authors aim to provide evidence, in vertebrate cells, for a phenomenon known in bacteria and lower eukaryotes as “untargeted” or “hitchhiking” mutations. The effects of translesion synthesis (TLS) polymerases and PCNA ubiquitination are also studied, in line with the idea that mutations downstream from the lesion must result from TLS. The analysis is technically sound, and the paper is well written for the most part. This reviewer, however, feels that the study is more appropriate for a specialized journal. The concept of multiple mutations has been extensively studied. While the analysis confirms their occurrence in chicken and human cells, it provides limited information on the mechanisms of these mutations.

Major comments:

1) It is generally known that mutagenic treatments induce closely spaced mutations. While the authors argue that studies in vertebrate cells are limited, the genome-wide analysis presented here is complicated due to the inaccuracy of predicting primary lesion sites (see below). The insight into the mechanisms of multiple mutations from this study is also limited due to a lack of adequate approaches (also see below). Thus, the conclusions hardly go beyond re-stating that closely spaced mutations exist.

2) The lesion position is deduced from the type of mutation and the identity of neighboring nucleotides. This is a major limitation reducing the power of the analysis. As a start, what is the accuracy of defining “primary” mutations? Can one deduce from previous studies what percentage of mutations at true (known) lesions matches the characteristic "primary" mutation type and context? How is the inaccuracy of the prediction accounted for in the analysis of mutation pairs? Without these considerations, it is hard to determine which conclusions are supported by the data and which are not.

3) Interactions of DNA polymerases with DNA involve several nucleotides beyond the lesion site. The majority of “collateral” mutations in this study are found within this range. These events are part of the TLS process and do not inform on the extent of error-prone synthesis beyond the lesion. Deducing the “signature” of these collateral mutations is meaningless, as they are not distinct from the events at the lesion itself. A separate analysis is needed for the really close mutations and those further away from the “lesions” (e.g. > 6 nt). The authors acknowledge that close mutations result from the TLS itself in the Discussion, but this is not sufficient. This understanding needs to be incorporated into the overall design of the analysis.

4) The lack of studies in Pol zeta- and Rev1-deficient cells limits mechanistic insight. The severe sensitivity of these cells to the mutagenic treatments indicates that these polymerases play the primary role in the bypass of the respective lesions. While some effects of Pol kappa and Pol eta on very close mutation pairs are observed, these effects are likely secondary and difficult to interpret in the absence of more comprehensive studies of the bypass mechanism. Perhaps analysis of Pol zeta/Rev1-deficient cells and combinations of polymerase defects at lower doses of cisplatin and UVC could provide more insight.

5) Many statements in the text are not supported by statistical analysis of the data.

Minor comments:

1) The first paragraph on p. 3: reference 16 also studied chromosomal UV-induced DNA lesions.

2) Legend to Figure 1, panel A should be revised to avoid the impression that the model was proposed in this work.

3) What are “consecutive mutations” in Figure 3C?

**Have all data underlying the figures and results presented in the manuscript been provided?**

Reviewer #1: Yes

Reviewer #2: **No: **The sequencing data is deposited in a public repository and the called mutations are provided in the supplement. However, the numerical data the underlies graphs is only provided for the NMF calculations. To recapitulate the other graphs, someone would need to recount all the corresponding mutations from the mutation list and preform downstream calculations.

Reviewer #3: Yes

PLOS authors have the option to publish the peer review history of their article (what does this mean?). If published, this will include your full peer review and any attached files.

Reviewer #1: No

Reviewer #2: No

Reviewer #3: No

---

## [Decision Letter · Decision Letter 1]

12 Jan 2022

Dear Dr Szüts,

Thank you very much for submitting your Research Article entitled 'Characterisation of the spectrum and genetic dependence of collateral mutations induced by translesion DNA synthesis' to PLOS Genetics.

The manuscript was fully evaluated at the editorial level and by independent peer reviewers. The reviewers appreciated the attention to an important topic but reviewer 2 identified concerns that we ask you address in a revised manuscript.  Please also fulfill the requirement of data availability policy about providing source numerical data as reminded by reviewer 2 and outlined in this letter below.

We therefore ask you to modify the manuscript according to the review recommendations. Your revisions should address the specific points made by each reviewer.

[LINK]

Yours sincerely,

Dmitry A. Gordenin, Ph.D.

Associate Editor

PLOS Genetics

Gregory P. Copenhaver

Editor-in-Chief

PLOS Genetics

Reviewer's Responses to Questions

**Comments to the Authors:**

Reviewer #1: .

Reviewer #2: The authors have now sufficiently addressed all of my previous minor concerns. They declined to assess the genetic dependencies of collateral mutagenesis due to UVC light, stating that this would be interesting, but too expensive to conduct the additional sequencing. While needing to avoid increased expense is understandable, it does miss an opportunity to increase the novelty and impact of the work. They additionally state that REV1-/- and REV3L-/- genotypes could not be evaluated due the low dose of cisplatin required to maintain viability would result in very few mutations, which is a reasonable explanation. The clarification of the calculation for the percent of non-primary mutations attributable to collateral mutagenesis is very helpful and supports ~12% of these mutations being caused by this mechanism. The authors also now provide analysis to validate the use of NMF to identify a collateral mutagenesis signature.

One point, however, was not well addressed: the comparison of close mutation ratios instead of the total number of mutation ratios per genome. The authors argue that the use of ratios is more intuitive because cisplatin treatment produces similar numbers of close mutation pairs, but lesser overall mutations as UVC treatment and therefore the difference in spectrum would not be as apparent. While this is true for comparing overall mutation spectra between damaging agents, the use of close mutation ratios to assess differences between different genotypes treated with the same damaging agent is problematic because both the total mutation load and the number of close mutation pairs could be impacted by the change in genotype. This makes it difficult to know if a reduction in close mutation ratio is due to a loss of close mutation pairs or an increase in primary mutations. These analyses should be done for the authors to state their genotypes are specifically impacting the formation of close mutations.

Reviewer #3: The prior criticisms have been addressed satisfactorily.

**Have all data underlying the figures and results presented in the manuscript been provided?**

Reviewer #1: Yes

Reviewer #2: **No: **It does not appear that the numerical data that underlies graphs are provided in spreadsheet form as supporting information.

Reviewer #3: Yes

PLOS authors have the option to publish the peer review history of their article (what does this mean?). If published, this will include your full peer review and any attached files.

Reviewer #1: No

Reviewer #2: No

Reviewer #3: No

---

## [Editor Report · Decision Letter 2]

21 Jan 2022

Dear Dr Szüts,

We are pleased to inform you that your manuscript entitled "Characterisation of the spectrum and genetic dependence of collateral mutations induced by translesion DNA synthesis" has been editorially accepted for publication in PLOS Genetics. Congratulations!

Editors commend you for carefully addressing reviewers critique, performing additional work and submitting well organized source data. 

Yours sincerely,

Dmitry A. Gordenin, Ph.D.

Associate Editor

PLOS Genetics

Gregory P. Copenhaver

Editor-in-Chief

PLOS Genetics

Comments from the reviewers (if applicable):

**Data Deposition**

http://datadryad.org/submit?journalID=pgenetics&manu=PGENETICS-D-21-01339R2

**Press Queries**

---

## [Editor Report · Acceptance letter]

2 Feb 2022

PGENETICS-D-21-01339R2 

Characterisation of the spectrum and genetic dependence of collateral mutations induced by translesion DNA synthesis 

Dear Dr Szüts, 

We are pleased to inform you that your manuscript entitled "Characterisation of the spectrum and genetic dependence of collateral mutations induced by translesion DNA synthesis" has been formally accepted for publication in PLOS Genetics! Your manuscript is now with our production department and you will be notified of the publication date in due course.

With kind regards,

Katalin Szabo

PLOS Genetics

On behalf of:
